# Will the Interest Triggered by Virtual Reality (VR) Turn into Intention to Travel (VR vs. Corporeal)? The Moderating Effects of Customer Segmentation

**Lili Geng, Yufei Li and Yongji Xue ***

School of Economics & Management, Beijing Forestry University, Beijing 100083, China; lily0506@163.com (L.G.); Lee_yufei@163.com (Y.L.)

*   Correspondence: xyjbjfu@bjfu.edu.cn; Tel.: +86-010-6233-8447

**Abstract:** Under the new normal of COVID-19, interest in e-production/e-services has, increasingly, included Virtual Reality (VR) tourism. However, the relationship between the perceived need for VR tourism and the stimulation of intention to corporeal tourism is, yet, vague, where corporeal tourism refers to visiting actual tourism destinations. To investigate the preferred intention of particular tourist modes (VR vs. corporeal), an integrated framework was proposed, by merging key elements from the attention, interest, desire, and action (AIDA) model and the technology-acceptance model (TAM). A sample of 657 respondents was collected, during February 2022, and hypotheses were tested using a partial least square structural equation model (PLS-SEM). The findings showed that interest in VR tourism had a strong hierarchical effect on the behavioral intention to a particular tourist mode, mediated by perceived usefulness or ease of use, attitude, and desire. Interest was significantly linked to two key constructs of TAM, whilst both determined attitude. Attitude significantly influenced the preference intention toward a particular tourism mode, directly and indirectly with users' desires, as a crucial mediator in the relationship. The individual characteristics moderate the paths, from evaluation to attitude and attitude to the mediator of desire to intention. This study contributes to both theories as well as practices in tourism management and marketing.

**Keywords:** AIDA model; VR tourism; corporeal tourism; PLS-SEM; customer segmentation

## 1. Introduction

The outbreak of COVID-19 brought enormous pressure and challenges to the tourism industry. Under the concept of new normal, coined by the World Travel and Tourism Council (WTTC, 2020) [1], integrated-digital-identity solutions and contactless travel are fast-growing [2]. Virtual museums, cloud tourism, and virtual places of interest have sprung up [3]. Virtual reality (VR) tourism is believed to contribute to building the resilience of the tourism industry, when facing emergencies and fulfillment of some consumers' travel desires [4]. VR tourism, also, provides a relatively rich tourism experience for people with economic constraints, limited time, and/or poor health [5].

Virtual reality (VR) is defined as a collection of interactive 3D technologies that provide synthetic feedback to one or more senses, immersing users in a world beyond reality [6]. The concept of VR tourism, in this study, refers to creating or reproducing a virtual-tourism environment, based on real scenes, through the digital collection and storage of tourism resources and the 3D visual simulation of tourism landscapes, to provide an immersive experience and real-time interaction [7]. Under the context of the progress of information and communication technologies (ICTs) as well as the COVID-19 crisis, VR tourism is attracting an increasing number of people, who want to gain authentic experiences and hedonic values [8,9]. Whether VR tourism can promote or, even, replace corporeal tourism in the post-COVID-19 era has received significant research attention. However, research on the relationship between the perceived need or experience of VR tourism and the intention

for corporeal tourism remains insufficient, and consumers' decision-making processes are crucial to the recovery and sustainable development of tourism in the post-epidemic era, both of which need further discussion.

VR tourism and corporeal tourism both have their advantages. Previous researchers pointed out that through the promotion and advocacy of VR tourism only, the benefits tourists hope to obtain through corporeal tourism will be damaged [10]. VR serves as a pre-experience marketing tool, to relieve tourists' misunderstanding about the destination, by stimulating the sense of interest, immersion, and enjoyment in the pre-experience stage and providing an important reference for tourism decision- and strategy-making [11,12]. Tourists' preference for a particular mode of tourism is not fixed [13]. Instead of debating a particular tourism mode, this study suggests tourism managers and marketers should consider the potential factors that affect the tourists' preference intention toward a particular tourism mode (VR vs. corporeal). Considering that preference behavior varies, based on different circumstances and conditions [14,15], this study aims to clarify the factors influencing tourism-mode preference-behavior, by including the intention toward VR and corporeal tourism, in one theoretical framework.

Interest is an important form and the most active component of motivation. The existing research indicated that if users expect to get more information about the destination experienced in VR [16], then interest and desire will lead to a behavioral intention to visit. Data from Statista (https://www.statista.com/, accessed on 27 March 2022), a global statistical database, indicated that around 70% of consumers ages 19–49 are very or quite interested in VR [17]. Several studies confirmed the quantity association between visitors' interest in VR applications and their desire or willingness to be involved [11,18]. However, how the vital factor, interest, transforms into the behavior intention toward VR vs. corporeal tourism, has not been fully revealed.

In terms of technology-based products or services, it may be difficult to foresee the behavior processes in which interest touched off [19]. In this regard, the technology-acceptance model (TAM) is a well-established and influential theory that seeks to explain how users accept technology [20]. Whether or not to go to the corporeal destination that is recommended by VR tourism depends on the internal drive of travelers, to a large extent [14]. According to the attention, interest, desire, and action (AIDA) model [21], emotional responses play a primary role in travel-behavior outcomes. However, keeping tourists' interest and making them perceive the benefit of a particular service, to reach the next stage of the model, is considered the most difficult [22]. Thus, taking interest, as the starting point, to explore how these kinds of emotional reactions affect subsequent behavior intention, is important.

Users' intentions toward a particular tourism mode differ, due to different environmental conditions and user groups [13]. For example, due to mobility barriers, most elderly people prefer to improve their quality of life and well-being through the VR tourism mode [23]. Previous studies suggest that gender, age, education level, income, and other demographic characteristics affect people's preference for a specific mode [8]. The individual characteristics that may moderate the decision paths of particular tourism modes are more worthy of discussion than the substitution of VR for the corporeal tourism mode [24].

To bridge this research gap, an integrated AIDA and TAM theoretical framework is established in this study, to explore the underlying process of the tourists' preference intention toward specific tourism modes, triggered by an interest in VR. Secondly, it focuses on moderating the effects of gender, experience using VR, and age (three classes). This study makes several theoretical and practical contributions: it helps explain consumer-preference behavior intention to a particular tourism mode (VR vs. corporeal), by identifying the factors of interest, evaluation (perceived usefulness, perceived ease of use), attitude, and desire to VR tourism, based on a model integrated from the two theories. It responds to calls for examining user-group characteristics, as a vital boundary condition in the decision path of a particular tourism model [8,13,25]. The findings provide tourism administrators and practitioners potentially helpful insights into marketing and promotion of VR tourism

under the new normal and inspirations for the sustainable and healthy development of corporeal tourism in the post-COVID-19 era.

The remaining structure of this study is presented below. The theoretical background and hypotheses development are discussed in Section 2. The research data and methodology are presented in Section 3. The empirical results are covered in Section 4. Section 5 shows the discussion of outcomes and implications. In the end, conclusions and future research are summarized in Section 6.

## 2. Theoretical Background and Hypotheses

### 2.1. Integration of Theories

This study integrates the TAM and AIDA model, to examine the factors that influence consumers' preference behavioral intentions toward VR vs. corporeal tourism [20]. TAM has made outstanding theoretical contributions to understanding technology acceptance and application. The initial purpose of TAM is to explain the determinant factors of technology acceptance. TAM investigates the user's adoption of technology or computer systems, as a further extension of the theory of reasoned action (TRA). The actual use behavior is driven by behavioral intent, which is led by attitude toward use. Further, the two particular beliefs, perceived usefulness and perceived ease of use, influence attitude. Parsimony is a significant advantage of the TAM model, but its explanatory power to predict behavior intention ignores several other factors and shows defects [26]. In the context of this study, the nature of consumers should be considered [27], such as psychological reactions [19].

The AIDA model was, originally, used to predict the effectiveness of advertising communication or media marketing, and, now, is widely cited in related disciplines. The model was built on the premise that customers go through a sequence of stages, from cognitive to emotive to behavioral. There are four stages that help marketers grasp how targeted customers react to advertisements or propaganda information as well as change their psychological state and consumption behavior over time. As AIDA proposed, interest and desire are important emotional determinants of behavioral intention. Previous research demonstrated that interest and adoption or use intention were positively correlated [28]. Desire, which serves as the motivation to adopt or purchase a product or service, has been proven to be a powerful factor to explain behavior intention [29,30].

This study developed an integrated model of AIDA and TAM, in a bid to cover their insufficient and give full play to advantages, to investigate how the interest in VR tourism affects consumers' preference behavior intention to specific tourism modes, and what role attitude and desire play in this process.

### 2.2. Hypotheses and Conceptual Framework

2.2.1. Interest, Perceived Usefulness, and Ease of Use

Interest in VR tourism refers to people's curiosity or interest aroused by external information and media advertising [31]. However, the process from interest to whether they actually use the mode of VR tourism is affected by a series of factors. People go through internal psychological changes to attitudes, intention and achieve actual action. Several hierarchical factors have been added to the model, such as perception, consciousness, evaluation, knowledge, cognition, understanding and so on [32,33].

In VR tourism, an item of technology-based products or services, when predicting the behavioral preference with interest as the pre-variable, the factors of technical threshold should be considered due to its complexity [19]. A potential user not disturbed by technical anxiety or discomfort in the decision-making process will obtain benefits from using it. Their attitude toward VR tourism will be stimulated [34]. In the effect evaluation of tourism advertising, the AIDA model is extended to the AIDEA by adding perceived usefulness and perceived credibility [33], in view of the peculiarity of tourism services compared with general goods or products. Interest has been proven to have a significant positive impact on perceived usefulness [9,34]. Some literatures revealed the flow experience delivered by

the virtual tourism experience affects tourists' tendencies to use mediated by perceived usefulness [35]. The users' interest triggered by information from surroundings, impacts the perceived usefulness of the delivery app [19]. People who are interested in a specific application or equipment tend to collect more information, which increases perceived ease of use. Thus, the following hypotheses are proposed:

**Hypothesis 1 (H1).** *Interest in virtual reality tourism (INT) has a positive impact on the perceived usefulness (PU) of VR tourism.*

**Hypothesis 2 (H2).** *Interest in virtual reality tourism (INT) has a positive impact on the perceived ease of use (PEU) of VR tourism.*

2.2.2. Perceived Usefulness, Perceived Ease of Use, and Attitude

Perceived usefulness reflects the degree to which a person thinks that using a specific system will improve their performance [20]. The perceived ease of use reflects how easy a person thinks it is to use a specific system. In the context of preference intention to VR tourism, Perceived usefulness refers to the level that a user feels that VR tourism is useful for improving their utility, including enjoyable experience, convenient and efficient to achieve tourism purposes, and so on. Perceived ease of use is defined as the extent to which a person believes that the utilization of VR tourism is easy and without efforts. VR tourism devices are supposed to be simple to use, easy to operate, and provide clear information provided.

Attitude is how the user assesses the preferability of applying VR tourism. Research has revealed the perceived usefulness for E-production can lead to greater preferences attitude for technical format of production or service. When people perceive a low level of effort use a particular technology, the more likely they are to have a positive attitude toward using it [20,36,37]. The strong link of perceived usefulness or perceived ease of use and attitude has been proven by large numbers of scholars [38,39]. It is reasonable to believe that users are more likely to hold a positive attitude, when they think using VR tourism can improve their performance or utility (PU), without too much effort (PEOU). Hence, the following hypotheses are proposed:

**Hypothesis 3 (H3).** *Perceived usefulness has a positive impact on attitude toward VR tourism.*

**Hypothesis 4 (H4).** *Perceived ease of use has a positive impact on attitude toward VR tourism.*

2.2.3. Attitude and Desire

Attitude refers to how a person views or evaluates the behavior or object they are interested in, either positively or negatively. The AIDA theory posits that interest is the stage that motivates customer emotion [21]. After interest is aroused, an individual's wish to obtain or enjoy a product or service emerges, which is the desire. The model of goal-directed behavior (MGB) claimed that desire was a nonnegligible mediator on the relationship between attitude and intention [30], that is, attitude has a positive impact on desire. To turn interest into desire and, then, transform into intention, potential users' interest or enthusiasm and positive attitude need to be maintained.

The above evidence demonstrates that interest does not alter desire directly, but does so indirectly, through attitude. The former results revealed that the attitude toward new technology served as a mediator in the link of interest and desire [19]. Attitude is the antecedent of motivation, which is the primary cognitive factor that determines desire and intention [40]. That is, when consumers hold a positive attitude toward VR tourism, the desire to utilize it will increase over time [41]. The following hypothesis is developed:

**Hypothesis 5 (H5).** *Attitude toward VR tourism has a positive impact on the desire to use VR tourism.*

### 2.2.4. Attitude, Desire, and Behavioral Intention

Behavioral intention (or preference-behavior intention) refers to the probability that customers like to use VR vs. the corporeal tourism mode. In the field of acceptance of tourism-related VR, attitude has a significant impact on intention [42–44]. However, how/when the positive attitude turns into the preferred intention to the VR tourism mode is uncertain, and corporeal tourism may become the best alternative. The intention to choose a certain travel mode is largely affected by the attitude about it [41]. If a specific service fails to meet customers' usefulness expectations, it may lead to their negative attitude toward it, so they will no longer employ the technology or service. The favorite attitude toward VR tourism will result in intention to use almost being widely recognized [38,43]. Simultaneously, the satisfaction of the immersive and memorable experience or mental imagery obtained through VR tourism will further enhance their intention to travel to the same destination on-site [24,45]. The development of VR tourism is immature, and technical discomfort or technical anxiety may disturb users, when utilizing and operating the VR equipment; as tourism information may be recorded by online platforms, there are concerns about privacy and security [35]. The negative attitude toward VR tourism will promote people's preference for the corporeal tourism mode. Therefore, the following hypotheses are proposed:

**Hypothesis 6 (H6).** *Attitude toward VR tourism has a positive impact on the intention to use VR tourism.*

**Hypothesis 7 (H7).** *Attitude toward VR tourism has a positive impact on the intention to go on corporeal tourism.*

An individuals' desire refers to an inner impulsion for a goal or purpose that propels them to engage in a behavior [40]. Studies emphasized the mediating effect of desire to travel on an ecologically friendly airline in the enhancement of eco-friendly travel intentions [46]. Travel attitudes, according to [41], have a significant impact on tourist mode choice, via the mediating effects of desire. Earlier studies tested the strong link between desire and behavioral intention, in terms of technology-based products or services [47]. Desire is a process that precedes intention in the psychological decision-making process, so it is worth noting the difference between the two. Intention, usually, requires a certain degree of self-efficacy, relatively clear commitment, and planning, while desire does not. Tourism is regarded as a high-level product, to meet people's non-material cultural needs. There are multiple interference factors between desire and intention, which can lead to different intentions toward specific tourism modes. Distinguished from desire, intention can be disturbed by factors, such as perceived behavioral control or the limitations of VR equipment conditions. Based on sufficient conditions, travelers driven by desire will be more bound to apply VR tourism. Studies argue that most people have feelings and attachment to tourism destinations, and they can obtain the fun and enjoyment that virtual tourism cannot achieve through corporeal tourism, such as intimate interaction with friends [48], contact with natural landscape [49,50], and experience of food, culture, and other entertainment projects [51]. The desire of VR tourism will stimulate people's intention to travel on-site. The following hypotheses were formulated:

**Hypothesis 8 (H8).** *Desire to use VR tourism has a positive impact on intention to use VR tourism.*

**Hypothesis 9 (H9).** *Desire to use VR tourism has a positive impact on intention to engage in corporeal tourism.*

2.2.5. Moderated Mediation Effect of Gender, VR Usage Experience, and Age

Predictors will not work equally well for the behavior outcomes of all consumers. The moderating effect of age, gender, and experience is likely to strengthen or weaken the intention to employ VR applications in the tourism sector and participate in corporeal tourism [5]. Existing studies examined whether gender, age, and experience variables had significant moderating effects, on the relationships between behavior intention and other latent variables [52–54]. Experience in using specific systems or products can increase users' comfort and familiarity, in continuing to use [23]. In marketing, it is commonplace to determine market segments by generation, such as baby boomers and millennials. Similarly, the tourism industry is no exception [45]. Previous studies have shown that different genders have a different attachment to tourism destinations and the value perception of experience [13]. Therefore, the following hypotheses are proposed:

**Hypothesis 9 (H9a–i).** *Gender (GN), VR usage experience (EXP), and age (AG) have moderating effects on the nine hypothetical relationships proposed above.*

The proposed research model of this study is shown in Figure 1, below.

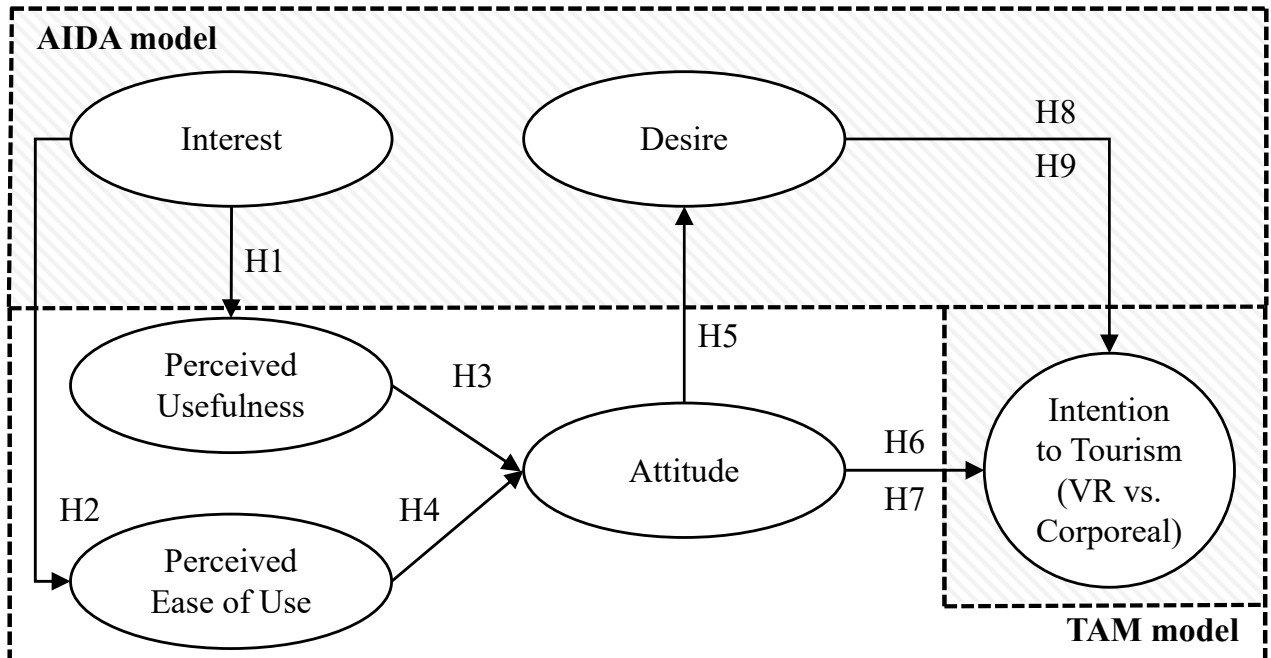

**Figure 1.** Proposed research model.

### 3. Methodology

*3.1. Sample and Data Collection*

After the pilot survey in January 2022, the final data were from a formal investigation executed by the research team, from February to March 2022. Numerous anonymous online questionnaires were delivered through the QUESTIONSTAR platform. To ensure the validity of the questionnaire, a brief explanation of the concepts of VR tourism and corporeal tourism is provided, to help respondents have a better understanding. We inserted a link to VR tourism and set the minimum response time. To avoid common-method bias (CMB), a screening question and a duplication question were asked.

The target survey areas and objects were chosen by random and stratified sampling and distributed in five provinces and cities in China: Beijing, Shanghai, Guangzhou, Zhejiang, and Jiangsu. A total of 160 questionnaires were distributed in each city, and 683 complete answers were collected from 800 respondents. Of these, 5, in which respondent's answer time was less than 180 s, and 21, where they responded in repetitive patterns,

were eliminated. Hence, 657 replies were kept for analysis, an effective feedback rate of 82.12%.

### 3.2. Measurement

The measurement scale was derived from previous studies and was revised to the context of VR applications in tourism. As per the theoretical hypotheses, there are seven constructs to be assessed, each structure corresponds to 3–5 items, respectively. In the AIDA model: interest and desire were adopted from the work of [55,56]. To measure the core variables of TAM, including perceived usefulness, perceived ease of use, attitude, and intention to VR or corporeal tourism, a scale provided by [20] was adapted. To use terms familiar to the respondents and express them clearly and concisely as far as possible, after translating the English items into Chinese, the expression was modified. A pilot test, with 30 respondents that participated, was conducted before the formal survey, to guaranteed the reliability of the questionnaire. The items were refined further, according to their feedback. The questionnaire items of the research constructs are shown in Appendix A.

A seven-point Likert scale, ranging from "1, strongly disagree to 7, strongly agree" was used. Gender was measured using dummy variables, "1" for "men" and "0" for "women". Experience of VR tourism usage was measured as a binary variable, with "1" indicating "have previous VR tourism usage experience" and, otherwise, "0". Age was classified into four groups: 16–25, 26–35, 36–50, and above 50 (see Table 1).

**Table 1.** Demographic profile of the sample (*n* = 657).

| Characteristics | Frequency | Percentage | Variables | Frequency | Percentage |
|---|---|---|---|---|---|
| Gender | | | Age | | |
| Male | 340 | 51.80 | 16–25 | 106 | 16.10 |
| Female | 317 | 48.20 | 26–35 | 410 | 62.40 |
| Occupation | | | 36–50 | 129 | 19.63 |
| Enterprise employee | 469 | 71.40 | >50 | 12 | 1.83 |
| Technical personnel | 62 | 9.50 | Education | | |
| Freelance | 27 | 4.10 | High school or below | 43 | 6.50 |
| Government official | 59 | 9.00 | Associate or Bachelor's degree | 557 | 84.80 |
| Student | 34 | 5.20 | Master's degree or above | 57 | 8.70 |
| Other | 6 | 0.90 | Healthy | | |
| Personal monthly Income (RMB) | | | Very healthy | 114 | 17.40 |
| Less than 4000 | 60 | 9.13 | Healthy | 279 | 42.50 |
| 4001–6000 | 88 | 13.39 | General | 189 | 28.80 |
| 6001–8000 | 155 | 23.59 | Less healthy | 66 | 10.00 |
| 8001–10,000 | 159 | 24.20 | Unhealthy | 9 | 1.40 |
| More than 15,000 | 195 | 29.68 | | | |

### 3.3. Common Method Bias (CMB)

Data were tested for validity, including no response bias, CMB, and multicollinearity as well as reliability and validity. To check the existence of nonresponse bias, Chi-squared tests, together with independent sample *t*-tests, were executed, following the approach widely used. Disparity between the initial and the end 220 responses was compared. Results of the Chi-squared and *t*-tests revealed no significant differences between the two selected groups ($p < 0.05$), demonstrating the non-response bias is non-existent.

To estimate the potential CMB, Harman's single-factor test was applied [57]. According to the suggestions and the report results, the number of characteristic root factors greater than 1 was more than one, and the variance interpretation rate of the first factor before rotation was 37.33% < 50%, proving common method variance was not a critical issue [57]. The results of the complete collinearity test indicated that the pathological VIF levels of all potential structures range from 1.249 to 1.523 [58], less than the threshold of 3.3, inferring that CMB was not a major concern. It is criticized that the single-factor

and collinearity tests are no longer acceptable [59]. We, further, adopt a technology in common construction methods, comparing the fitting index of the model that loads all the indicators of interest into the corresponding factors with that of one that includes a single method factor [57,60]. As CFA outcomes showed the $\Delta$ RMSEA $= 0.01 < 0.05$, $\Delta$ RSMR $= 0.007 < 0.05$, $\Delta$ CFI $= 0.023 < 0.1$, and $\Delta$ TLI $= 0.024 < 0.1$ [61], meaning that there is no problem with CMB.

### 3.4. Methods

We tested the hypotheses using Smart PLS version 3.0 and analyzed the data collected, based on the nature of the model and samples. In the Smart PLS software, PLS-SEM can perform a hypothesis verifying potential and difficult measurement variables. Research proposes that PLS-SEM requires two phases [62]. Firstly, we need to evaluate the measurement model, including the construct reliability (internal consistency), convergent and discriminant validity for the indicators of latent variables, and the fitting of the proposed structural framework. To do so, the outer loading, Cronbach's $\alpha$, composite reliability (CR), and average variance extracted (AVE) were screened. We tested the discriminant validity of the scale, by examining whether the square root of AVE of each construct was larger than the correlation with other factors, and whether the HTMT (heterotrait-monotrait) ratios of correlation are less than 0.85 [63,64].

The second stage is the hypothesis-path test, which, also, includes the mediation-effect test. We tested the moderating effect of gender and VR usage experience on every hypothesized relationship in the structural model, using multi-group analysis (MGA) or between-group analysis [65]. To analyze the moderating effect of age in more detail, we divided the sample into three groups, specifically, 16–25 years old ($n = 106$), 26–35 years old ($n = 410$), and 36–50 years old ($n = 129$). Since only 12 respondents were over 50, they were excluded. Considering that age is divided into three categories, and the research model is a chain-mediated model, pairwise comparison for group differences is more complex, so the moderated-mediation effect was tested using the SPSS 22.0 Macro-Process (IBM, New York, NY, USA).

## 4. Results

### 4.1. Sample Demographic Analysis

The demographic statistics are reported in Table 1. Among them, 51.80% of the respondents were male, and the remaining 48.20% were female. For occupation, 71.40% of the respondents are enterprise employees, 9.50% are technical personnel, 4.10% are freelancers, 9.00% are government officials, 5.20% are students, and the remaining 0.90% are other occupations. Regarding personal monthly income, 9.13% are paid less than RMB 4000, 13.39% RMB 4001–6000, 23.59% RMB 6001–8000, 24.20% RMB 8001–10,000, and 29.68% had a monthly income of more than RMB 15,000. In terms of age, 16.10% of the respondents are 16–25 years old, 62.40% are 26–35 years old, 19.63 % are 36–50 years old, and 1.83% are above 50 years old. With regard to education, 6.50% of the respondents had a high school education or below, 80.84% had an associate or bachelor's degree, and 8.7% hold a master's degree and above. Among the respondents, 17.40% thought they were very healthy, 42.50% thought they were in a healthy state, 28.80% thought they were in a generally healthy state, 10.00% thought they were less healthy, and only 1.40% thought they were in an unhealthy state.

### 4.2. Measurement Model Evaluation

SPSS 22.0 was used to explore the reliability, convergent, and discriminant validity of all the items and constructs. As shown in Table 2, the AVE and outer loadings exceeded the limit value of 0.50 and 0.70, so the convergent validity was confirmed. The reliability was determined because the CR value is greater than 0.8. All Cronbach's alpha values were close to the threshold 0.7. The rho_A values were all near the threshold of 0.7. These reflected that the internal consistency and reliability are acceptable [66]. Regarding the

discriminant validity, the square root of AVE in each component was larger than the other correlation values among the constructs (see Table 3). Table 4 indicates that all HTMT ratios were less than 0.85 and suggested that the measurement model has discriminant validity [64].

**Table 2.** Construct validity.

| Construct | Item | Outer Loadings | T-Values | AVE | CR | Cronbach's $\alpha$ | rho_A |
|---|---|---|---|---|---|---|---|
| DES | DES1 | 0.797 | 45.018 | 0.587 | 0.850 | 0.767 | 0.775 |
| | DES2 | 0.778 | 47.207 | | | | |
| | DES3 | 0.728 | 25.555 | | | | |
| | DES4 | 0.761 | 29.963 | | | | |
| ATT | ATT1 | 0.750 | 31.616 | 0.614 | 0.827 | 0.723 | 0.686 |
| | ATT2 | 0.804 | 48.860 | | | | |
| | ATT3 | 0.795 | 49.046 | | | | |
| PU | PU1 | 0.783 | 35.525 | 0.605 | 0.822 | 0.703 | 0.674 |
| | PU2 | 0.790 | 41.488 | | | | |
| | PU3 | 0.761 | 31.675 | | | | |
| INT | INT1 | 0.887 | 59.950 | 0.782 | 0.878 | 0.721 | 0.721 |
| | INT2 | 0.882 | 64.184 | | | | |
| PEU | PEU1 | 0.721 | 22.813 | 0.582 | 0.807 | 0.684 | 0.652 |
| | PEU2 | 0.781 | 36.241 | | | | |
| | PEU3 | 0.786 | 31.322 | | | | |
| VIR | VIR1 | 0.857 | 58.895 | 0.734 | 0.846 | 0.686 | 0.637 |
| | VIR2 | 0.856 | 52.075 | | | | |
| COR | COR1 | 0.835 | 62.672 | 0.617 | 0.828 | 0.691 | 0.705 |
| | COR2 | 0.764 | 29.460 | | | | |
| | COR3 | 0.756 | 28.940 | | | | |

Note: Average variance extracted (AVE), composite reliability (CR). All the values of AVE, CR, Cronbach's alpha, and rho A are significant at the $p < 0.001$ level.

**Table 3.** Discriminate validity of research model.

| | AVE | ATT | DES | INT | PEU | PU | COR | VIR |
|---|---|---|---|---|---|---|---|---|
| ATT | 0.614 | 0.783 | | | | | | |
| DES | 0.587 | 0.553 | 0.766 | | | | | |
| INT | 0.782 | 0.499 | 0.527 | 0.884 | | | | |
| PEU | 0.582 | 0.525 | 0.478 | 0.402 | 0.763 | | | |
| PU | 0.605 | 0.589 | 0.471 | 0.538 | 0.551 | 0.778 | | |
| COR | 0.617 | 0.528 | 0.531 | 0.480 | 0.568 | 0.565 | 0.786 | |
| VIR | 0.734 | 0.540 | 0.566 | 0.534 | 0.465 | 0.534 | 0.494 | 0.857 |

**Table 4.** HTMT ratio.

| HTMT | ATT | DES | INT | PEU | PU | COR | VIR |
|---|---|---|---|---|---|---|---|
| ATT | | | | | | | |
| DES | 0.749 | | | | | | |
| INT | 0.709 | 0.700 | | | | | |
| PEU | 0.783 | 0.658 | 0.581 | | | | |
| PU | 0.867 | 0.642 | 0.770 | 0.826 | | | |
| COR | 0.763 | 0.714 | 0.678 | 0.842 | 0.828 | | |
| VIR | 0.819 | 0.803 | 0.787 | 0.715 | 0.815 | 0.732 | |

Notes: HTMT should be lower than 0.85.

*4.3. Structural Model Assessment*

The structural model was evaluated using bootstrapping, with bias-corrected and accelerated (BCa) confidence intervals, with 5000 sub-samples and two-tailed significance at 0.05 levels. The path relationships, coefficients, significance, and explanatory variances

($R^2$) are presented in Table 5. The structural model has no problem of collinearity because the VIF value was less than 3.3. If the value of adjusted $R^2$ coefficient of determination or the variance is closer to 1, the prediction ability of the determinants will be more powerful. Followed by the suggestions, the $R^2$ value for predictability larger than 0.67 is considered substantial, 0.33 is medium, and 0.19 is considered weak [67]. The results show that almost 40% ($R^2$ = 0.395) of the intention to VR tourism can be disclosed, by the latent constructs that compose it, and 36.1% of intention to corporeal tourism is explained ($R^2$ = 0.361). The prediction ability of antecedent constructs to attitude is moderate, as the $R^2$ value is 0.405. This means that more than 40.5% of attitude can be explained by perceived ease of use, along with perceived usefulness. A total of 30.6% of variance in desire is explained by attitude ($R^2$ = 0.306). The size of the $f^2$ effect was tested, to offset the possible deficiency of evaluating the explanation accuracy through the R-value [68]. Most of the $f^2$ values are at a high or medium level, greater than 0.35 and 0.15, respectively, confirming that the PLS path model has good prediction accuracy. All the $Q^2$ values obtained by the blindfolding technique are greater than zero, achieving the prediction relevance of the model.

**Table 5.** Assessment of the structural model.

| Paths | Beta | S.E. | T-Values | *p*-Values | BC Boot 95% CI LLCI | BC Boot 95% CI ULCI | Support | VIF | $R^2$ | $f^2$ | $Q^2$ |
|---|---|---|---|---|---|---|---|---|---|---|---|
| H1: INT → PU | 0.538 | 0.031 | 17.108 | 0 | 0.479 | 0.6 | Yes | 1.000 | 0.289 (PU) | 0.406 | 0.172 (PU) |
| H2: INT → PEU | 0.402 | 0.033 | 12.212 | 0 | 0.34 | 0.471 | Yes | 1.000 | 0.162 (PEU) | 0.193 | 0.091 (PEU) |
| H3: PU → ATT | 0.431 | 0.04 | 10.813 | 0 | 0.345 | 0.5 | Yes | 1.435 | 0.405 (ATT) | 0.218 | 0.245 (ATT) |
| H4: PEU → ATT | 0.287 | 0.044 | 6.495 | 0 | 0.201 | 0.374 | Yes | 1.435 | | 0.097 | |
| H5: ATT → DES | 0.553 | 0.029 | 19.284 | 0 | 0.497 | 0.612 | Yes | 1.000 | 0.306 (DES) | 0.441 | 0.172 (DES) |
| H6: ATT → VIR | 0.328 | 0.042 | 7.886 | 0 | 0.237 | 0.405 | Yes | 1.441 | 0.395 (VIR) | 0.123 | 0.286 (VIR) |
| H7: ATT → COR | 0.337 | 0.05 | 6.739 | 0 | 0.235 | 0.43 | Yes | 1.441 | 0.361 (COR) | 0.124 | 0.218 (COR) |
| H8: DES → VIR | 0.385 | 0.045 | 8.464 | 0 | 0.294 | 0.474 | Yes | 1.441 | | 0.170 | |
| H9: DES → COR | 0.345 | 0.052 | 6.693 | 0 | 0.248 | 0.455 | Yes | 1.441 | | 0.129 | |

The test results of the path hypothesis (H1–H9) of the conceptual model are shown in Table 5 and Figure 2. Interest in VR tourism affects perceived usefulness (H1: β = 0.538, t = 17.108, *p* < 0.001) and perceived ease of use (H2: β = 0.402, t = 12.212, *p* < 0.001), supporting H1 and H2. The direct links among perceived usefulness (H3: β = 0.431, t = 10.813, *p* < 0.001), perceived ease of use (H4: β = 0.385, t = 8.464, *p* < 0.001), and attitude is significant, supporting H3 and H4. Attitude affects the desire significantly (H5: β = 0.553, t = 19.284, *p* < 0.001), supporting H5. We, further, studied the contacts among attitude, desire, and behavioral intention. We discovered support for H6 and H7, where the attitude significantly affects intention to VR tourism (H6: β = 0.328, t = 7.886, *p* < 0.001) and intention to corporeal tourism (H7: β = 0.337, t = 6.739, *p* < 0.001). We found support for desire significantly affects intention to corporeal tourism (H8: β = 0.345, t = 6.693, *p* < 0.001) and intention to VR tourism (H9: β = 0.385, t = 8.464, *p* < 0.001).

To test the indirect effect of the hypothetical paths, the bootstrapping method was applied, which ensures appropriate results in measuring the confidence interval of indirect relationships. The results showed that interest has an indirect influence on attitude through perceived usefulness and ease of use (β = 0.347, LLCI = 0.295, ULCI = 0.397, *p* < 0.001). Since the indirect effect of attitude on intention to VR tourism (β = 0.540, LLCI = 0.471, ULCI = 0.591, *p* < 0.001) or corporeal tourism (β = 0.528, LLCI = 0.464, ULCI = 0.595, *p* < 0.001) is significant, desire is confirmed to be a mediator in this relationship. The multiple mediating effects of interest on travel intention to particular tourism mode are, also, significant (β = 0.188, LLCI = 0.145, ULCI = 0.225, *p* <0.001; β =0.183, LLCI = 0.149, ULCI = 0.223, *p* < 0.001), in the presence of perceived usefulness or perceived ease of use, attitude and desire are simultaneous or non-simultaneous, respectively. Together with the specific indirect effects, the results are presented in Table 6.

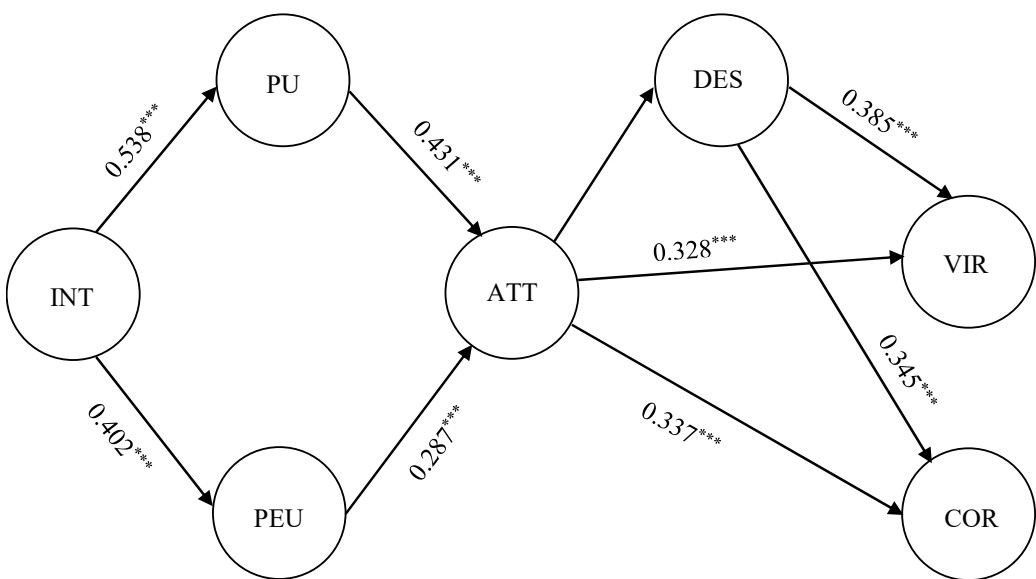

**Figure 2.** Path analysis results. Note: ***: *p* < 0.001.

**Table 6.** Result of mediating effects.

| | Coefficient (β) | T Statistics | *p*-Values | 95% BC CI |
|---|---|---|---|---|
| | Total Indirect effects | | | |
| INT → ATT | 0.347 | 13.652 | 0.000 | 0.295; 0.397 |
| ATT → VIR | 0.540 | 15.584 | 0.000 | 0.471; 0.591 |
| ATT → COR | 0.528 | 17.367 | 0.000 | 0.464; 0.595 |
| INT → VIR | 0.188 | 9.119 | 0.000 | 0.145; 0.225 |
| INT → COR | 0.183 | 9.116 | 0.000 | 0.149; 0.223 |
| | Specific Indirect effects | | | |
| INT → PEU → ATT → VIR | 0.038 | 4.018 | 0.000 | 0.023; 0.059 |
| INT → PU → ATT → VIR | 0.076 | 5.094 | 0.000 | 0.049; 0.106 |
| INT → PEU → ATT → DES → VIR | 0.025 | 4.471 | 0.000 | 0.015; 0.036 |
| INT → PU → ATT → DES → VIR | 0.049 | 5.715 | 0.000 | 0.034; 0.066 |
| INT → PEU → ATT → COR | 0.039 | 3.719 | 0.000 | 0.020; 0.063 |
| INT → PU → ATT → COR | 0.078 | 5.532 | 0.000 | 0.054; 0.107 |
| INT → PEU → ATT → DES → COR | 0.022 | 4.403 | 0.000 | 0.014; 0.033 |
| INT → PU → ATT → DES → COR | 0.044 | 5.104 | 0.000 | 0.028; 0.064 |

*4.4. Group Difference Testing*

This study examined whether gender and experience (with or without) varies in the hypothetical path for preference behavior to a particular tourism mode. The overall samples were divided into two groups, according to gender (male = 340, female = 317) and experience in using VR tourism (experience = 312, no experience = 345), respectively. The PLS-MGA results are reported in Table 7. A significant difference between groups is derived from *p*-values equal to or less than 0.05 and equal to or more than 0.950. The gender differences, between the attitude and intention to VR tourism (H6, *p*-value 0.040) and the desire and intention to VR tourism (H8, *p*-value 0.007), are significant. The moderating effects of experience, for influences of perceived usefulness on attitude (H3, *p*-value 0.025), and perceived ease of use (H4, *p*-value 0.008) on attitude, are evident. However, there was no significant difference for gender and experience in other paths.

The moderating role of age, in the indirect effects of the interest in VR on attitude through evaluation (perceived usefulness and perceived ease of use) and attitude, on the behavior preference of the specific tourism mode through desire, was examined. This is known as a moderated mediation model. Within the bias-corrected 95% confidence interval, if there is no 0-value between the lower and upper bounds of the confidence interval, the

moderated mediating effect hypothesis is considered accepted. Table 8 reported the results of the conditional indirect effects of attitude on the intention to engage in VR or corporeal tourism, via desire pertaining to the stage of age. The results indicated the conditional indirect effects of attitude on the intention to use VR tourism were significant for the 16–25 age group ($\beta$ =0.2936; 95% CI = [0.1850, 0.4130]), the 26–35 age group ($\beta$ = 0.2093; 95% CI = [0.1507, 0.2725]), and the 36–50 age group ($\beta$ = 0.1024; 95% CI = [0.0013, 0.2017]). The conditional indirect effects of attitude on intention to engage in corporeal tourism were, also, significant for the 16–25 age group ($\beta$ =0.1707; 95% CI = [0.0484, 0.3199]), the 26–35 age group ($\beta$ = 0.2224; 95% CI = [0.1558, 0.2928]), and the 36–50 age group ($\beta$ = 0.1726; 95% CI = [0.0595, 0.2912]). However, there was no significant moderated mediating effect for the other paths.

**Table 7.** MGA results.

| Parameters | Path Coefficients | | | MGA | Remark | Path Coefficients | | | MGA | Remark |
|---|---|---|---|---|---|---|---|---|---|---|
| | Male (M) | Female (F) | Diff. | $p$-Value M vs. F | | Exp (E) | No-Exp (NE) | Diff | $p$-Value E vs. NE | |
| H1:INT → PU | 0.529 | 0.545 | −0.016 | 0.801 | Not | 0.314 | 0.363 | −0.059 | 0.353 | Not |
| H2:INT → PEU | 0.354 | 0.455 | −0.101 | 0.142 | Not | 0.166 | 0.404 | −0.108 | 0.118 | Not |
| H3: PU → ATT | 0.464 | 0.401 | 0.063 | 0.433 | Not | 0.349 | 0.293 | 0.181 | 0.025 | Supported |
| H4: PEU → ATT | 0.29 | 0.282 | 0.007 | 0.932 | Not | 0.522 | 0.341 | −0.238 | 0.008 | Supported |
| H5: ATT → DES | 0.532 | 0.58 | −0.048 | 0.399 | Not | 0.515 | 0.593 | −0.078 | 0.175 | Not |
| H6: ATT → VIR | 0.408 | 0.237 | 0.17 | 0.040 | Supported | 0.349 | 0.457 | 0.056 | 0.542 | Not |
| H7: ATT → COR | 0.335 | 0.34 | −0.005 | 0.948 | Not | 0.507 | 0.565 | −0.174 | 0.055 | Not |
| H8: DES → VIR | 0.271 | 0.513 | −0.241 | 0.007 | Supported | 0.252 | 0.426 | −0.075 | 0.424 | Not |
| H9: DES → COR | 0.292 | 0.398 | −0.105 | 0.261 | Not | 0.351 | 0.426 | −0.049 | 0.582 | Not |

**Table 8.** Conditional indirect effects of attitude on intention of virtual vs. corporeal tourism modes, through desire moderated by age.

| Items | Virtual Reality Tourism | | | | Corporeal Tourism | | | |
|---|---|---|---|---|---|---|---|---|
| Group by Age | | | Boot 95% CI | | | | Boot 95% CI | |
| | Effect | Boot SE | LLCI | ULCI | Effect | Boot SE | LLCI | ULCI |
| 16–25 | 0.2936 | 0.0577 | 0.1850 | 0.4130 | 0.1707 | 0.0712 | 0.0484 | 0.3199 |
| 26–35 | 0.2093 | 0.0310 | 0.1507 | 0.2725 | 0.2224 | 0.0349 | 0.1558 | 0.2928 |
| 36–50 | 0.1024 | 0.0506 | 0.0013 | 0.2017 | 0.1726 | 0.0593 | 0.0595 | 0.2912 |

Notes: bootstrap sample size = 5000, 95% CI = 95% confidence interval, LLCI = lower limit of confidence interval, ULCI = upper limit of confidence interval.

Further, through pairwise comparison, the differences of moderate effects among the three age groups were analyzed. Table 9 represented the results of differences between conditional indirect effects on the relationship between attitude and intention toward VR tourism. Results in link of attitude toward intention toward corporeal tourism were omitted, due to the fact they were not significant. The index is the effect of pairwise contrast, for example, W1 refers to the subtractive difference of conditional indirect effects between the 26–35-year-old group and the 16–25-year-old group. There were significant differences in the conditional indirect effects between the 36–50 age group and the 16–25 age group (index = −0.1912, LLCI = −0.3483, ULCI =−0.0453). There was no significant difference between the other two groups (W1 and W3). To further demonstrate the moderating effect of the differences among ages, the moderated mediating effects were plotted (see Figure 3). The plotted marginal effects indicated the younger the age is, the greater the positive moderated effect of desire on VR tourism intention (see Figure 3a). Compared with the other two groups, for the youngest group, the moderated effect on the relationship of attitude and VR tourism intention is the smallest (see Figure 3b).

**Table 9.** Index of moderated mediation (difference between conditional indirect effects).

| Pairwise Contrasts | Index | Boot SE | Boot 95% CI | |
|---|---|---|---|---|
| | | | LLCI | ULCI |
| W1 | −0.0843 | 0.0618 | −0.2094 | 0.034 |
| W2 | −0.1912 | 0.0768 | −0.3482 | −0.0453 |
| W3 | −0.1069 | 0.0598 | −0.2325 | 0.0069 |

Note: The 16–25 age group is the benchmark group and the conditional indirect effects are defined as Eff1, with Eff2 for the 26–35 age group and Eff3 for the 36–50 age group. W1 = Eff2 minus Eff1; W2 = Eff3 minus Eff1; W3 = Eff3 minus Eff2.

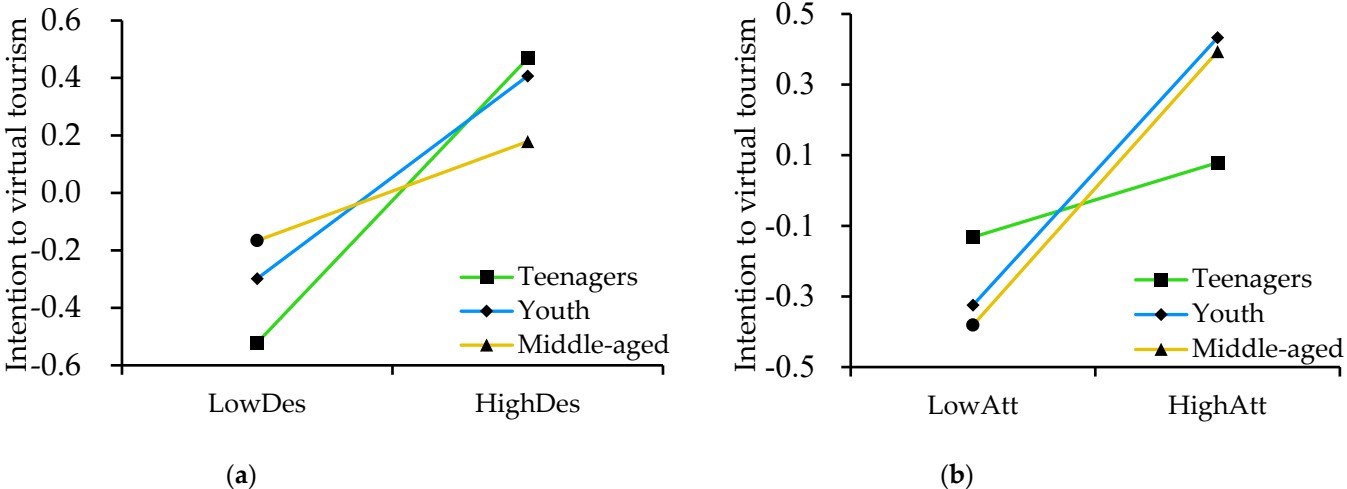

**Figure 3.** (**a**): the conditional effects of desire on intention to virtual tourism moderated by age. (**b**): the conditional effects of attitude on intention to virtual tourism moderated by age. Note: teenagers: 16–25; youth: 26–35; middle-aged: 36–50.

## 5. Discussion

This study investigated the preferred intention to particular tourist modes (VR vs. corporeal), based on a sample of 657 respondents and the PLS-SEM approach. Differences were found among user segmentation (gender, VR usage experience, and age) in the hypothesis paths. All hypothetical paths in the research model were supported (H1–H9), and the hypotheses of moderating effects were partially supported (H9a–i).

Interest in VR tourism is a powerful predictor of consumers' use of specific tourism models and is in line with the related research [19,22]. Another author found that the type of destination and advertising had no significant impact on stimulating tourists' interest [25], suggesting the effect of interest variables on evaluating tourists' tourism behavior is, still, vague. The current study gives a partial response for it, revealing that interest in VR tourism has positive effects on tourists' behavior intention to travel (virtual vs. actual), indirectly through the role of a series of mediators, such as evaluation, attitude, and desire. Interest was found to be an important antecedent to perceived usefulness and perceived ease of use. In other words, people interested in VR tourism tend to think it is useful, and the technology is easy to operate. Cognitive overload, namely, the amount of mental processing power needed to use a particular tourism mode, can cause customers to leave and not complete their desired tasks [69]. The research results strengthened the strong connection between the antecedent variables of AIDA and the core variables in TAM [19,70].

In the case of the variables of attitude, they play a key mediating role in the links of technical cognitive factors (perceived usefulness and perceived ease of use) and emotional motivation factors (desire). VR tourism is a kind of technology-based product or service, while perceived usefulness and perceived ease of use are important antecedents of attitude, so the same consequence is, also, gained through many related research topics [42,44,71]. At the same time, the indirect effect of perceived usefulness is greater than perceived ease

of use, which is, also, a successful dialogue with research on technology adoption or VR application [9,44]. It, further, proves that the usefulness of products or services is necessary, in the acceptance of VR tourism.

Desire is found to have a significant positive impact on users' preference behavior for specific tourism modes. Previous research revealed a mediation effect of desire, between attitude and behavioral intention, in different contexts [72]. Results displayed that for the influence path of interest on behavior intention, whether for VR or corporeal tourism, the indirect effect of desire as a mediating factor is smaller than that of when it is not, which in line with the previous research [33]. It implies that for tourism, as a high-level product to meet people's consumption demand, the impact of desire on the final actual behavior is unstable, and it is easy to change via the used environment and conditions. Optimistic attitude and desire toward VR tourism, interactive experience between humanity and nature, happiness in interpersonal communication in the process of tourism, and so on may significantly influence the user's preference behavior, in a particular tourism mode [49,51,73]. The results explained why some people still prefer to travel to a physical destination, despite the growing interest in and convenience of using VR tourism [74].

The moderating effects of gender, VR usage experience, and age on the relationship of the selected constructs explaining the endogenous variables of behavior intention toward a specific tourism mode and an interest in VR tourism were investigated. For men, the attitude toward VR tourism has a greater impact on the behavior preference of the VR tourism mode than women. For women, if the desire to use VR tourism is sufficiently strong, the possibility of it transforming into intention is greater than men. There was an argument that implied that women are more inclined to show overall satisfaction with creative tourism than men, which provides a support for the conclusion in this study [75]. It, also, confirms that in VR tourism marketing, stimulating women's desire can promote the consumption outcomes more effectively. For customers who have not used VR tourism, the positive impact of perceived usefulness and perceived ease of use on attitude is limited, compared with the experienced tourist, in line with view in [70]. This result sheds light on developing customers without VR tourism experience, under the restriction of environmental conditions [47]. With the increase in age, the influence of attitude on intention to VR tourism gradually weakens under the mediation of desire. The same result was, also, gained in a similar study [74], which argued that older people had more positive attitudes toward the purchase intention of video rather than VR devices, but students were more likely to accept such devices. It shows that the marketing strategy of tourism products, by stimulating impulse consumption, is more effective for younger groups. This result, also, suggested that customers ages 26–50 may be restricted by more factors, in the process of transforming desire into behavior intention.

## 6. Implications and Limitations

### 6.1. Theoretical Implications

This study makes academic contributions through the integration of the TAM and AIDA model, by adding perceived usefulness and ease of use to the latter, as evaluation dimensions, and giving full play to their respective superiority. It suggested a holistic framework related to the decision-making process from interest to behavioral intention to a specific tourism mode. This study adds to the scant literature that applies the AIDA model to tourism, thereby, theoretically, enriching and innovating this discipline [5]. The interest in VR tourism, which is, also, a novel perspective, has a significant positive effect on willingness toward corporeal travel. This study proves that people's preference behavior for a particular tourism mode is, mostly, determined by the moderators of perceived usefulness—attitude and confirms the view that meeting the customers' utilitarian needs for tourism content is a key aspect in tourism marketing [4,9].

The dependent variable is expanded into two dimensions: the behavioral intention of VR vs the corporeal tourism mode. To the best of our knowledge, few scholars have integrated both tourism forms into the same theoretical framework. This study focused

on and confirmed the positive impact of the interest, evaluation, and attitude toward virtual tourism, on intention to engage in corporeal tourism, which could be regarded as an enrichment of the existing models or literature, examining the factors influencing behavioral intention to VR tourism or real tourism, separately [8,76]. This study provides a reference for further understanding the trade-off and synergetic process of users, regarding the two tourism modes.

The important mediating role of attitude and desire in the integration model is tested, to clarify the explanation mechanism of the relationship between interest in VR tourism and preference behavior for a specific tourism mode. This study, empirically, proved that attitude develops prior to the shapes of desire. It has a positive effect on the intention to use a specific tourism model through the mediator of desire, directly and indirectly. This result confirmed the results of peer research and expands the application of the MGB viewpoint, in the field of tourism [29,30]. VR tourism is the integration of VR technology and traditional tourism. When investigating tourists' preference behavior for specific travel modes, the cognition or perception of technology itself will, inevitably, affect the users' experience of the tourism process. This study reveals the positive mediating effect of perceived usefulness and perceived ease of use as perceived variables, and desire as an emotional motivation variable in the decision-making process, which has not been studied in this field, to date. The outcome enlightened the research on the acceptance of the combination of products comprising new technology, so experience or entertainment products and services should consider both extrinsic perceptions and intrinsic motivations.

*6.2. Management Implications*

This study has several management implications for tourism marketers and VR developers or designers. The moderating impacts of group characteristics, namely, gender, experience, and age, on the relationships between interest in VR tourism and behavior intention to a particular tourism mode (VR vs. corporeal), offer insights into potentially successful marketing practices. Building interest, maintaining it, and transforming it into desire are considered the most difficult stages, to promote actual consumption behavior, according to AIDA. Given that the results have shown that perceived usefulness and perceived ease of use have a positive impact on positive attitudes, it is vital to emphasize the benefits and simplicity of using VR tourism, to hold the interest of potential users. From a tourism-marketer perspective, to make users feel the utility of VR tourism, it is necessary to expose that it can bring happiness and well-being to tourists, via the publicity information. If tourism marketers update the visual landscape and 3D interactive experience settings of corresponding destinations in VR tourism equipment in real time, according to the changes of seasons and weather, the authentic experience and perceived usefulness of tourists will be improved effectively and arouse the intention to corporeal tourism [77]. Tourism marketers can benefit from these measures, whether from the realization of the independent value of VR tourism or the increase in corporeal tourism consumption.

Desire significantly affects tourists' preference behavior for specific tourism modes. However, desire is a relatively subjective concept, and its impact on actual tourism behavior is unstable and limited by conditions. Marketers should seek to enhance the authentic experience, immersion, presence, and interaction of VR tourism [44,47,76], which are of great assistance in holding customers' desire, tending to go further toward corporeal tourism. During the COVID-19 pandemic, it is necessary to provide inspiration or encouragement to customers via various of social media, such as Facebook, YouTube, Weibo, etc., in an interactive way, to stimulate their intentions toward VR travel. As for precise marketing on VR tourism in gendered market segments, marketers should promote men's VR tourism behavior intention, by promoting toward them to form a positive attitude. When the target group is women, efforts to improve their desire will be more effective. The 16–25 age group, usually, comprises a large proportion of students, whose desire and intention to use VR tourism could be fostered and enhanced effectively, through substantial price reduction

and promotion methods, such as discounts, group purchases, bundled sales, coupons, and so on.

The results, also, indicated that the innovators and firms that develop VR technology should not neglect their efforts, when developing convenient and useful VR tourism-related equipment. For VR tourism engineers, attention should be given to users' current technical literacy. That is, the VR tourism should be compatible with users' technical-operation habits or system experience. Enhancing users' technical comfort and reducing obstacles are helpful, to form a positive attitude toward VR tourism. Technology demonstrators should expound on how well the program will go, so users will not give up when they encounter difficulties during initial use. For example, a presentation video or animation is supposed to play to guide users step by step, when they first enter the VR tourism platform. A use experience with less effort, also, helps to enhance the willingness toward corporeal tourism. For prospective users who have no experience in VR tourism, focusing on improving their perception of the usefulness and ease of use will transform them into actual customers.

### 6.3. Limitations and Future Research

Strictly speaking, there is room for improvement of this study, given the following limitations. First, given that more than 70% of the respondents are company employees, and children and the elderly are not taken into account, this limits the generalizability. Second, this study was based on data collected within a single time frame, during the COVID-19 pandemic, when users' positive attitudes and tolerance toward VR tourism may be greater. Assuming that there was no epidemic, people may have different levels of interest as well as demand for VR tourism and corporeal tourism; whether such reasons will lead to any deviation from the results needs further investigation. For the study of tourists' preferences for specific tourism modes, special consideration should be given to the COVID-19 crisis, policy restrictions, and social limiting [78].

Valuable questions remain to be explored. This study has explored the differences of customers' group characteristics, in particular tourism-mode-behavior preferences, but further investigations are, also, welcomed and needed on the application environment of subdivided VR tourism and corporeal tourism. Full consideration and exploration of the impact of VR tourism on corporeal tourism and tourist decision-making, such as the different destination types (natural scenic spots and artificial scenic spots), degrees of immersion (no immersion, semi-immersion and full immersion), VR tourism devices (mobile phone app, smart glasses, VR head-mounted display, and wearable devices), periods (off-season and peak season), and distances (domestic and foreign), should be given.

### 7. Conclusions

This study proposed an integrated framework, by combining the core variables of the AIDA model and TAM, to understand the preferred intention toward particular tourism modes (VR vs. corporeal). PLS-SEM was adopted, based on a sample of 657 respondents. The result revealed that interest in VR tourism had a significant hierarchical effect on preference intention to a particular tourism mode, mediated by perceived usefulness or perceived ease of use, attitude, and desire. Interest was strongly linked with the key structure of TAM, whilst both models determined the attitude. Attitude significantly influenced the preferred intention toward a tourism mode, directly and indirectly, and users' desire was a crucial mediator in the relationship. The impact of interest on preference intention toward VR tourism was greater, due to the mediating effect of perceived usefulness, attitude, and desire. Interest was more linked to preference intention toward corporeal tourism, when only moderated by evaluation and attitude. From the perspective of market segmentation by individual characteristics, the impact of attitude on intention toward VR tourism is greater among men, but the impact of desire is greater among women. Second, for customers who have no experience with VR tourism usage, the positive impact of perceived usefulness and perceived ease of use on attitude is limited, compared to those with such experience.

Third, among the three age groups, with the increase in age, the influence of attitude on intention toward VR tourism gradually weakens under the mediation of desire.

**Author Contributions:** Conceptualization, L.G.; methodology, Y.L.; software, L.G.; validation, L.G. and Y.X.; formal analysis, L.G. and Y.L.; investigation, Y.L.; writing–original draft preparation, L.G. and Y.L.; writing–review and editing, Y.X., L.G. and Y.L.; visualization, L.G. and Y.L.; supervision, Y.X.; funding acquisition, Y.X. All authors have read and agreed to the published version of the manuscript.

**Funding:** This research was funded by the Fundamental Research Funds for the Central Universities, 2021SPS01 and 2021SRZ04.

**Institutional Review Board Statement:** Not applicable.

**Informed Consent Statement:** Not applicable.

**Data Availability Statement:** Data sharing is not applicable.

**Conflicts of Interest:** The authors declare no conflict of interest.

## Appendix A

**Table A1.** Questionnaire items.

| Constructs | Items | Sources |
|---|---|---|
| Interest (INT) | INT1: I become interested in the new form of virtual reality tourism.<br>INT2: I am very interested in virtual reality use in tourism. | Hudson et al., 2011 [55];<br>Lin and Huang, 2006 [56] |
| Perceived Ease of Use (PEU) | PEU1: It is easy for me to become skillful at using virtual reality tourism.<br>PEU2: My interaction with virtual reality tourism is clear and understandable.<br>PEU3: I think that use of virtual reality tourism is not complicated/does not require a lot of mental effort. | Davis et al., 1989 [20] |
| Perceived Useless (PU) | PU1: I think using virtual reality tourism will help me get more information about the destination.<br>PU2: I think using virtual reality tourism can improve the efficiency of travel.<br>PU3: I think it is very useful to use virtual reality tourism. | Davis et al., 1989 [20] |
| Attitude (ATT) | ATT1: Using virtual reality tourism is positive.<br>ATT2: Using virtual reality tourism is beneficial.<br>ATT3: Using virtual reality tourism is attractive. | Davis et al., 1989 [20] |
| Desire (DES) | DES1: My hope for virtual reality tourism is passionate.<br>DES2: I hope to use virtual reality tourism in the near future.<br>DES3: I want to use virtual reality tourism right now.<br>DES4: I have a positive feeling about using virtual reality tourism. | Hudson et al., 2011 [55];<br>Lin and Huang, 2006 [56] |
| Intention to virtual reality tourism (VIR) | VIR1: I am willing to use virtual reality tourism in the future.<br>VIR2: I will invest time and money in using virtual reality tourism in the future. | Davis et al., 1989 [20] |
| Intention to corporeal tourism (COR) | COR1: After virtual reality travel, I will try to go on corporeal tourism in the future.<br>COR2: After virtual reality travel, I will invest time and money in corporeal tourism in the future.<br>COR3: After virtual reality travel, I have the intention to participate in corporeal tourism activities. | Davis et al., 1989 [20] |

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
