# Peer review of "Will the Interest Triggered by Virtual Reality (VR) Turn into Intention to Travel (VR vs. Corporeal)? The Moderating Effects of Customer Segmentation"

_sustainability, doi:10.3390/su14127010_

Round 1

Reviewer 1 Report

Generalized comments

The typescript is too long. The overall structure needs some amendments. There are several phrases that have no sense, due to lacking one or more words, or due to inadequate wording. This problem should be corrected for consistence. Edition should be checked by a native speaker.

The referencing scheme adopted by authors is not commonly used. So I suggest adapting it accordingly. For instance, as an accepted norm, the ascending numeral attributed should appear between rectangular brackets [], and applied it throughout the main body text.

The first time that abbreviations are used – even in context – , they should come completely written. Readers may not be aware of what AR, AI or VR mean. Please correct accordingly.

Section 5 “Conclusion and Discussion” seems quite weird. Mostly these are separate sections and Discussion comes always before Conclusion(s).

Specific comments

Lines 31-2: Please add “to” to the sentence. “The outbreak of COVID-19 brought about enormous pressure and challenges to tourism industry.”

L 36: For instance “… and national parks 34, and virtual museums…” the number “34” does not make any sense in context. Probably, and this is my guess, given the literature inadequacy flaw, the authors were referring to references [3,4]. This is a bold and systematic error found across the entire manuscript. Please do fix this in the whole typescript.

L 47, 49, 62, 66, (…): Identical problem was identified. Please fix it accordingly. Use a comma “,” between both numerals.

L 70-2: The sentence reads awkward. Please rewrite it.

L 73-4: What is “Statista”? The sentence reads awkward. Please rewrite it.

Reviewer 2 Report

  • Line 489; what meant by the following table may be previous is accurate
  • Table 9 not mentioned in the text
  • Figure 1 not mentioned in the text
  • Figure 2 not mentioned in the text
  • It will be better if the references number are reduced
  • Line 436 and 437; H3 and H4 parameters must be rewritten in the same style of H1, H2 and H9 in the same paragraph.

Round 2

Reviewer 1 Report

As the article has improved considerably, possibly due to help from someone other than the authors, it should be considered in the acknowledgments to refer to that help. Please check how other authors do the acknowledgments in similar articles in this journal.

The abbreviation used should be AIDA, not ADIA. However, there are at least 3 times where ADIA appears. Please amend accordingly.

There are a few little sentences that don't make sense, even those that were added recently. Please improve the following sentences:

L 449-50: “However, there was no significant moderated mediating effect were fund in the other paths.” There is something wrong with this sentence. Please rewrite this sentence.

L 655: “… greater among men, but the impact of desire is greater among woman.” It should come “women” instead of “woman” in accordance with the plural “men” previously referred.
